# Pinging the brain with visual impulses reveals electrically active, not activity-silent, working memories

**Joao Barbosa**[1,2], **Diego Lozano-Soldevilla**[1,3,4], **Albert Compte**[1]*

**1** Institut d'Investigacions Biomèdiques August Pi i Sunyer (IDIBAPS), Barcelona, Spain, **2** Laboratoire de Neurosciences Cognitives et Computationnelles, Département d'Études Cognitives, École Normale Supérieure, PSL University, Paris, France, **3** Centro de Investigación Biomédica en Red de Enfermedades Raras (CIBERER), Madrid, Spain, **4** Laboratory for Clinical Neuroscience, Centre for Biomedical Technology, Universidad Politécnica de Madrid, Campus de Montegancedo, Pozuelo de Alarcón, Madrid, Spain

* acompte@clinic.cat

**Data Availability Statement:** All the code used for the analyses can be found at https://github.com/comptelab/reactivations. The data and data decoder scripts are available from the original

## Abstract

Persistently active neurons during mnemonic periods have been regarded as the mechanism underlying working memory maintenance. Alternatively, neuronal networks could instead store memories in fast synaptic changes, thus avoiding the biological cost of maintaining an active code through persistent neuronal firing. Such "activity-silent" codes have been proposed for specific conditions in which memories are maintained in a nonprioritized state, as for unattended but still relevant short-term memories. A hallmark of this "activity-silent" code is that these memories can be reactivated from silent, synaptic traces. Evidence for "activity-silent" working memory storage has come from human electroencephalography (EEG), in particular from the emergence of decodability (EEG reactivations) induced by visual impulses (termed pinging) during otherwise "silent" periods. Here, we reanalyze EEG data from such pinging studies. We find that the originally reported absence of memory decoding reflects weak statistical power, as decoding is possible based on more powered analyses or reanalysis using alpha power instead of raw voltage. This reveals that visual pinging EEG "reactivations" occur in the presence of an electrically active, not silent, code for unattended memories in these data. This crucial change in the evidence provided by this dataset prompts a reinterpretation of the mechanisms of EEG reactivations. We provide 2 possible explanations backed by computational models, and we discuss the relationship with TMS-induced EEG reactivations.

## Introduction

A hallmark of the activity-silent working memory framework [1] is that memories stored silently in synaptic traces through short-term synaptic plasticity can be reactivated through nonspecific stimuli [1–5]. Evidence supporting activity-silent working memory has recently emerged from human electroencephalography (EEG) [6,7], in particular from EEG reactivations of unattended memories induced by visual impulses [7]—the so-called visual pinging.

publications (Wolff et al., 2015 [18]; Wolff et al., 2017 [7]; Rose et al. 2016 [6]).

**Funding:** This work was funded by the Spanish Ministry of Science, Innovation and Universities and European Regional Development Fund (https://www.ciencia.gob.es/, Refs: BFU2015-65315-R and RTI2018-094190-B-I00) to JB, DLS, AC; by the Institute Carlos III, Spain (https://eng.isciii.es/eng.isciii.es/Paginas/Inicio.html, grant PIE 16/00014 to AC, DLS and grant AC20/00071 to AC; by the Cellex Foundation to DLS, AC; by the Generalitat de Catalunya (AGAUR, https://agaur.gencat.cat/en/inici/index.html, 2014SGR1265, 2017SGR01565) to JB, DLS, AC; and by the CERCA Programme/Generalitat de Catalunya (https://cerca.cat/en/) to AC. JB was supported by the Spanish Ministry of Economy and Competitiveness (FPI program) and by the Bial Foundation (https://www.bial.com/com/bial-foundation, Ref: 356/18). This work was developed at the building Centro Esther Koplowitz, Barcelona. The funders had no role in study design, data collection and analysis, decision to publish, or preparation of the manuscript.

**Competing interests:** The authors have declared that no competing interests exist.

Despite their relevance for upcoming memory-guided behavior, currently unattended memories could not be robustly decoded from raw EEG voltage traces [6,7] (Fig 1A, red). In view of this, unattended memories resemble memories rendered behaviorally irrelevant by a contextual cue (discarded, Fig 1B, dashed lines), but they differ from attended memories with similar upcoming behavioral requirements, which are represented in sustained, active codes [6–8] (Fig 1A and 1B, solid lines). This observation has been key in interpreting EEG reactivations in pinging studies as evidence for activity-silent storage (see, e.g., recent reviews [8–14] or [2–4,15] for explicit simulations of this interpretation of the data). Intriguingly, pinging-induced increase in EEG decodability occurred exclusively for items that remained relevant for future, memory-guided behavior, suggesting that only unattended but still relevant items were kept in activity-silent traces. The mechanisms for such selective reactivation of activity-silent traces are unclear, as in existing computational models of activity-silent storage [1,2,4,5,16,17] short-term plasticity changes are induced by neuronal activity, regardless of its behavioral relevance.

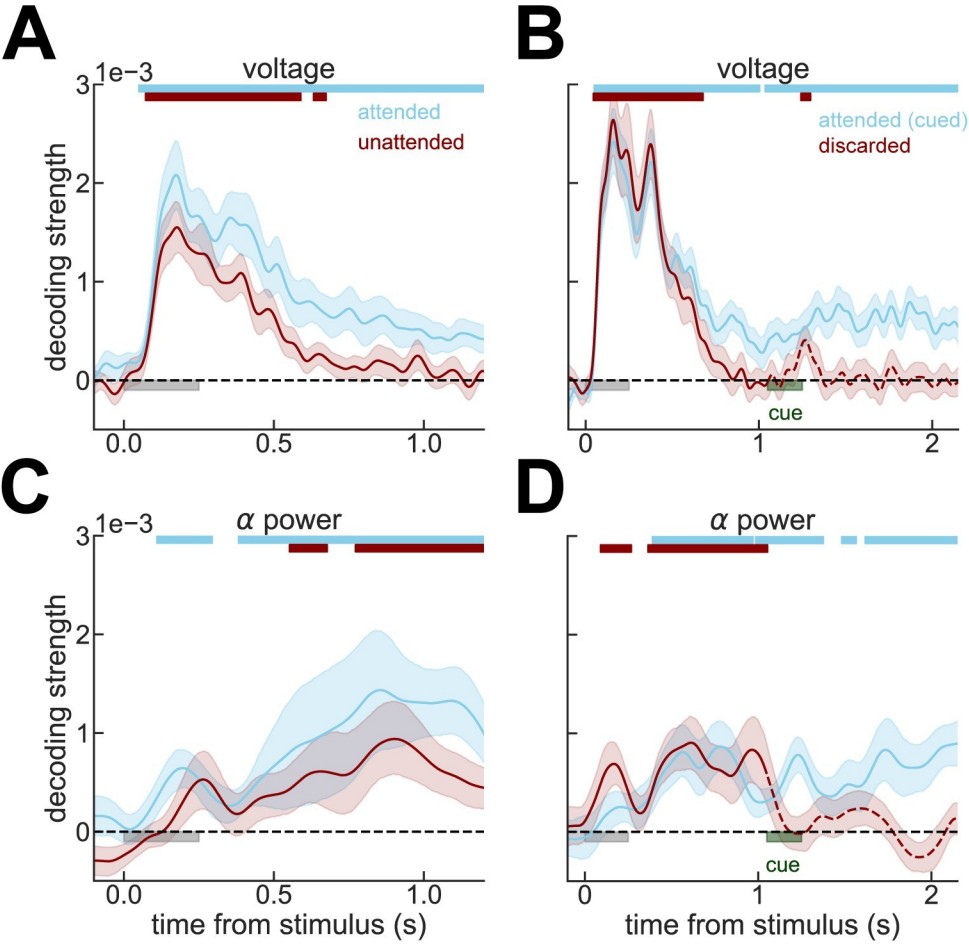

**Fig 1. Decoding from alpha power reveals an underlying, active working memory code. (A)** Strength of stimulus decoding from raw voltage traces as in [7]. As in the original study, unattended or **(B)** discarded memories cannot be decoded from raw voltage traces. **(C, D)** Same as (A) and (B), but decoding from alpha power (Methods), which reveals a sustained representation of the unattended stimulus. In (A) and (C), we analyze data from experiment 2 [7], while in (B) and (D), data from experiment 1 [7]. Light gray bars mark stimulus presentation periods. Notice that data immediately preceding pinging stimulus presentation are shown in this figure. Dashed lines mark the periods in which memories are irrelevant for upcoming behavior, following an instruction cue (dark green). All error bars are bootstrapped SEM, and color bars on the top mark the periods where bootstrapped 95% CI was above zero. Data from Wolff and colleagues (2017) [7].

Here, we reanalyzed EEG recordings of these influential pinging studies [7,18]. We found that unattended memories, previously shown to be inaccessible from scalp EEG voltages despite remaining behaviorally relevant [7], could in fact be decoded from raw EEG voltage and robustly from alpha power signals. This reanalysis demonstrates that the original claim of a silent representation of unattended memories is not supported by the data, which instead show an active code and thus calls for a reinterpretation of pinging-induced increases in EEG decodability. Finally, we argue that the increase in stimulus decodability following an unspecific stimulus, seen in human [6,7,18–20] and monkey electrophysiological experiments [21,22], can be explained by network models without short-term plasticity based on ongoing active, not silent, neural representations.

## Results

### Attended and unattended working memories are robustly decoded from alpha power

We realized that an earlier study had reported that (attended) spatial memories were decoded more reliably from EEG total alpha power than from evoked activity [23]. Thus, we analyzed alpha-power information content for attended, unattended, and no longer relevant orientation memories in the publicly available dataset of the original publication by Wolff and colleagues [7]. We found that a sustained alpha power code tracks the orientation of the items that remain relevant for future behavior, whether attended or unattended (Fig 1C and 1D, solid lines, S1 Fig). This shows that working memory contents are maintained in an electrically active neural code, even for items outside of the current attentional focus [24]. However, while attended memories were decodable both in alpha power and voltage traces, unattended memories were only robustly detected in alpha power. While this could reflect qualitative differences in what these 2 neural signals represent [25–27], we will explore here the parsimonious possibility that this stems from differential sensitivity of these 2 measures to the same underlying neural activity. Indeed, EEG voltage is known to lose decodability shortly following a reference baselining due to slow electrical drifts [28], while oscillations could be more robust to these baseline drifts. In this view, attended items would be represented by strong neural signals (represented both in voltage and in alpha power), while unattended items would be kept in analogous but weaker neural signals (picked only by alpha power). These neural dynamics could result from competitive interactions between prioritized and unprioritized memory items, in line with competing attractors in networks without activity-silent mechanisms [20,24,29].

### Lack of statistical power suggests spurious evidence for silent representations of unattended memories

In line with the hypothesis of an active but weaker representation of the unattended items, recent studies [30,31] show that lack of decodability for unattended working memories can be overcome by increasing statistical power (e.g., sample size). We wondered if a similar strategy could improve decodability of unattended orientations from raw voltages in this dataset. We addressed this tentatively in the current datasets with 2 complementary approaches. First, we attempted to increase the statistical power by (1) smoothing the voltage traces with a 32-ms kernel prior to decoding (instead of smoothing instantaneous decoding accuracies as done in the original study [7]; Methods); (2) averaging decoding accuracies over an interval of 200 ms before the pinging impulse; and (3) pooling trials from all sessions and subjects (Methods). We found that discarded memories were still not decodable from raw voltage ($p = 0.48$, t = −0.04, one-sided $t$ test), but the decoding of unattended memories almost reached the

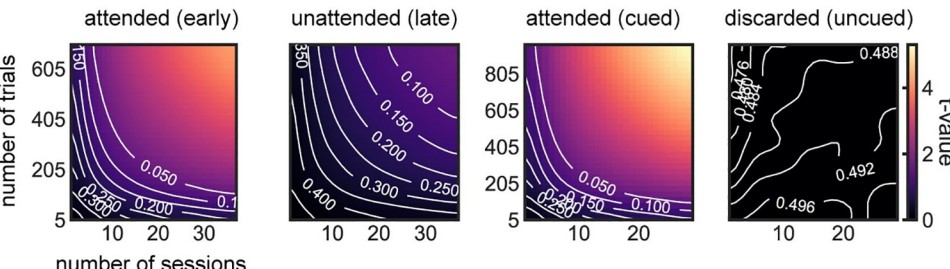

**Fig 2. Average voltage decoding during 200 ms prior to the impulse as a function of trial and session numbers** [37]. Attended items could be robustly decoded from voltage, but not discarded items. Decoding of unattended items suggested a possible underlying signal (*p* = 0.06, t = 1.55, one-sided *t* test, upper-right corner in second panel). Subsampling of sessions and trials was done by randomly subsampling (*n* = 5,000) without repetition from the full dataset (Methods). In white, contour lines for different *p*-value levels (one-sided *t* test). Data from Wolff and colleagues (2017) [7].

typical statistical threshold (*p* = 0.06, t = 1.55, one-sided *t* test). We also varied the number of sessions and trials included in the analyses to visualize how decoding depended on sample size (Fig 2). This further suggested that our analyses were underpowered and motivated our second approach, in which we reasoned that the low signal-to-noise ratio in this cohort could be due to specific sessions with overall low decodability. Sessions with low decodability could reflect technical issues during that particular session (e.g., EEG sensor placement) or specific subject characteristics, such as skull thickness or hair density. We thus divided our full dataset using cross-validation in high and low decoding sessions, based on the average decoding accuracy during the early delay ("split period" in Fig 3, Methods). We found that unattended memories could be robustly decoded during the whole delay (0.25–1.2 s, p = 0.002 randomization test, Methods) and in particular immediately before pinging (250 ms window, p = 0.039, randomization test, Methods) from high-decoding sessions, while discarded memories could not (both

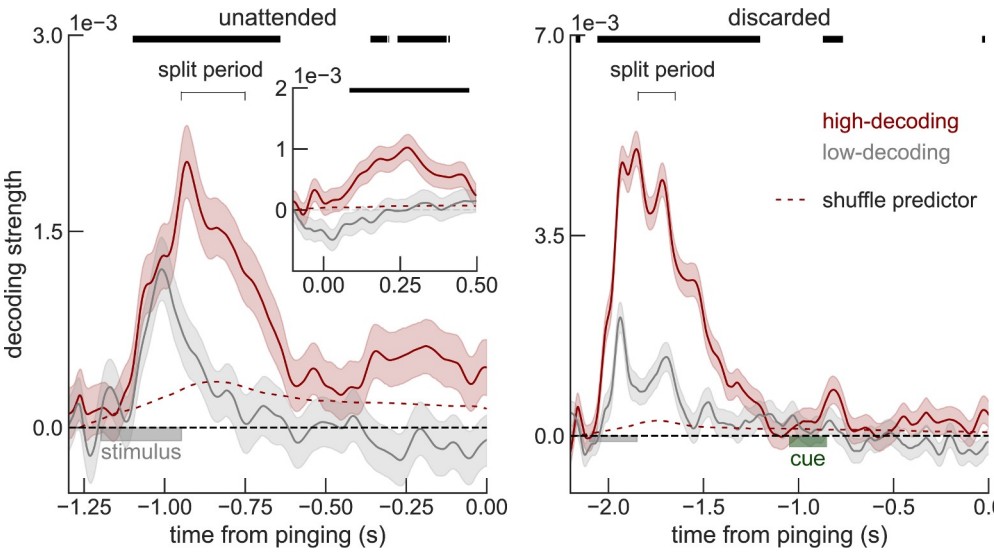

**Fig 3.** Sessions with high early-delay (split period, Methods) voltage decoding have a sustained code for unattended memories (left, red), but not for discarded memories (right). Replotting the reactivation period (inset), separately for high and low early-delay decoding sessions shows that "reactivations" only occur for sessions with a sustained code (left, red). Note that at time 0, the decoding strength is not actually zero (inset vs arrow). This is an artifact of baselining during the delay period (see also S3 Fig). Error bars are sem. Decoding strengths from high-decoding sessions were compared to the shuffle predictor (top black bars mark significant deviation, one-sided p<0.05, Methods). Time course and data are similar to Fig 1A and 1B. Data from Wolff and colleagues (2017) [7].

p>0.45, Fig 3). Finally, we show that sessions with high early-delay decoding ("split period" in Fig 3) are also those that have "reactivations" in voltage seen in the original publication (Fig 3, inset). Note that we used one-sided statistical tests (Figs 2 and 3), since negative decoding strengths are not expected. Additionally, one-sided statistical tests represent a conservative approach when claiming lack of decoding. These analyses, together with previous studies showing robust decoding of unattended memory items [22,30,32–36], suggest that also in this dataset, unattended items are not stored in activity-silent traces.

## Two plausible explanations for the increase in decodability that do not require activity-silent mechanisms

If there is an active EEG code for both attended and unattended stimuli prior to the visual impulse, as our analyses suggest, then what is the interpretation of the observed increase in EEG decodability [7,18]? We reasoned that EEG reactivation events may emerge from either an increase in the signal about the stimulus (as assumed in the activity-silent interpretation) or through a reduction in the across-trial variability (S3C Fig). In the data, we found that pinging reduces across-trial variability of EEG voltage (Fig 4A), as expected for neural responses to sensory stimuli [38]. In addition, we found that trials with stronger EEG decodability showed lower across-trial variability than trials with weaker EEG decodability during pinging (Fig 4B), demonstrating a link between trial-by-trial EEG variability and pinging-induced increase in EEG decodability. We argue that a reduction of variability with an otherwise intact active memory representation (Fig 3) is a parsimonious interpretation of the visual pinging effect. Alternatively, there is another interpretation of pinging-induced increases in EEG decodability consistent with our findings. Recent modeling work has shown how the enhancement of active representations [20,21] is expected when pinging recurrent neural networks with no need for activity-silent mechanisms [29,39]. An existing representation maintained in an attractor supported by recurrent and competitive interactions enhances its tuning when it is stimulated unspecifically (attractor-boost model, S2 Fig). Also, this mechanism would be consistent with these data, as it shows reduced variability (Fig 4C, gray), concomitant with boosted attractor tuning (S2 Fig). While both of these possible interpretations do not exclude an interplay of active representations with activity-silent mechanisms [5,17,40,41], they offer a parsimonious view that renders activity-silent working memory an inadequate framework to understand increases in decodability induced by nonspecific stimuli [7,18,20,21]. To further support this, we sought to evaluate variability predictions from a computational model where reactivations occur because of factual memory reactivation from silent, synaptic traces. We tested an available biophysical network model for (continuous) activity-silent working memory [5], which is an extension of the canonical but discrete model of activity-silent working memory [1] (Methods). In these simulations, a nonspecific input induced reactivations in some trials, causing an increase of across-trial variability (Fig 4C, black). This is because reactivations in such attractor networks are an all-or-none phenomenon, and great variability is expected when triggering them from weak, decaying activity-silent traces in noisy spiking networks. In sum, pinging reveals an underlying active memory, perhaps by reducing noise (Figs 4A, 4B, and S3C) in the presence of an active code (Figs 1 and 3) or by enhancing tuning in an active representation (S2 Fig), but not by reactivating stimulus signals from silent traces.

## Differential mechanisms for pinging- and TMS-induced reactivations

Increases of EEG decodability at the presentation of nonspecific impulses have been shown to occur not only for visual impulses [7] but also for external perturbation with single-pulse transcranial magnetic stimulation (TMS) [6]. Despite apparent similarities, TMS perturbations

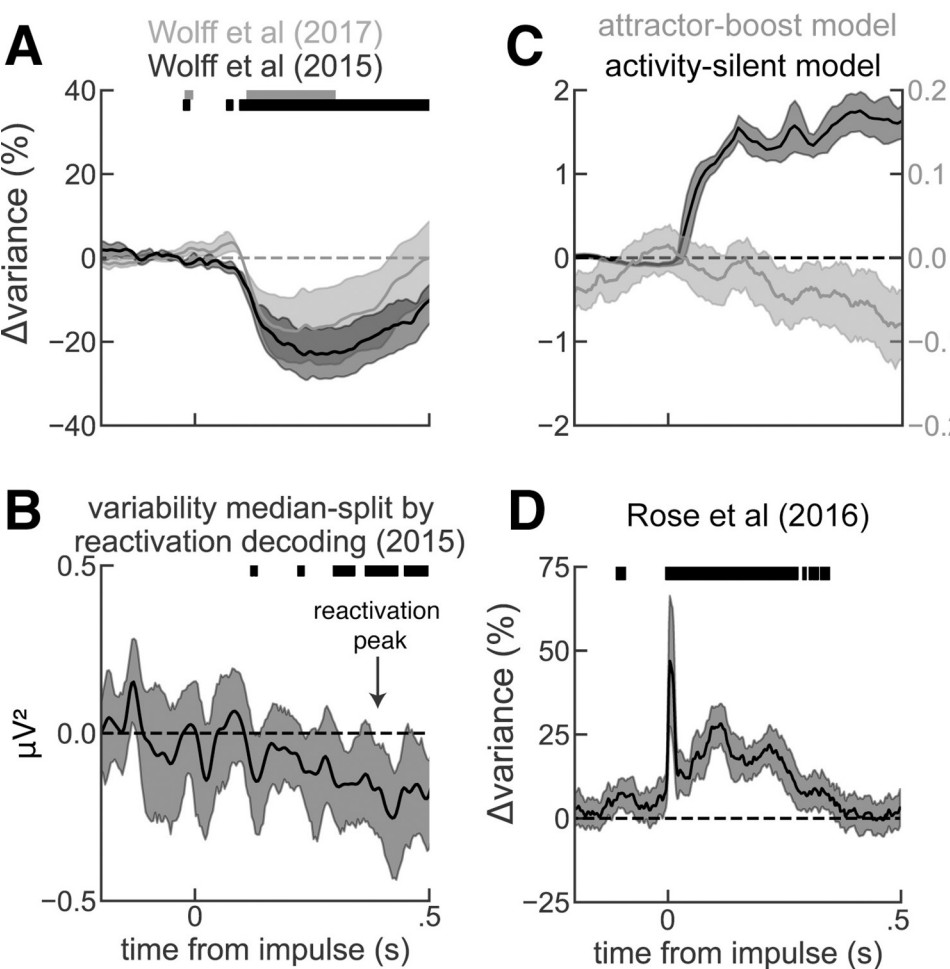

**Fig 4. Impulse-induced, across-trial variability change in human EEG and computational models.** (**A**) Percentage of variability change, relative to 0.2 s before the impulse, computed when the impulse was a visual stimulus in Wolff and colleagues (2017) [7] in gray and Wolff and colleagues (2015) [18] in black. (**B**) Difference in variances computed across trials with low vs high stimulus decoding (computed at the time of maximal decodability, black arrow) in Wolff and colleagues (2015) [18]. Trials with strong memory decoding showed significantly lower across-trial variance than trials with weak or absent memory decoding. We did not find a significant correlation in Wolff and colleagues (2017) [7] possibly because of a weaker pinging stimulus, which may have contributed to weaker increase in EEG decodability, not visible without baselining the data during the delay period (see S3 Fig). (**C**) Simulations of the activity-silent working memory model with short-term plasticity (dark) predict an increase of across-trial variability (Fano factor) following reactivations induced by a nonspecific drive. Simulations of the bump-attractor model without short-term plasticity (light) predict a decrease in the Fano factor following a nonspecific drive. See also S3 Fig. (**D**) Same as (A) but when the impulse was a single-pulse TMS (data from Rose and colleagues, 2016 [6]). Solid bars mark where the change in variability was significant (two-sided $t$ test, $p < 0.005$), and error bars are bootstrapped 95% CI of the mean. Six out of 54 sessions had outlier TMS artifact variance (i.e., extremely high variance at the time of the impulse) and were removed from this analysis. EEG, electroencephalography; TMS, transcranial magnetic stimulation.

impact working memory performance [5,6,42–44], while pinging does not [7,45] (see also chapter 4 of [46]). This suggested that these approaches could be interacting with fundamentally different neural mechanisms [7,47]. Indeed, we found through the reanalysis of the data of [6] that single-pulse TMS increases across-trial EEG variability (Fig 4D), in contrast to the reduction observed upon visual pinging (Fig 4A). Such increase in across-trial variability is in accordance with the activity-silent working memory model presented before (Fig 4C, black), thus potentially supporting the interpretation of TMS EEG reactivations as signals recovered

from activity-silent traces [5,6]. However, a note of caution is in order: The difficulty in precisely locating the TMS coil in different trials may contribute to increased EEG variability by virtue of the long-lasting effect of the TMS pulse on neural excitability [48]. This could mask EEG signals reflecting TMS-induced reactivations.

## Discussion

Through the reanalysis of existing datasets, we provide here evidence that working memory "reactivations" by visual pinging, considered prime evidence for activity-silent working memory [2–4,8–15], occur in the presence of active (not silent) ongoing memory representations in the delay period. In addition, we verify that representations for unattended items are notably weaker than for attended items, consistent with biased competition between active memories [29]. Based on this substrate for working memory, visual pinging may increase EEG decodability through (1) ping-induced reduction in across-trial variability; and/or (2) ping-induced boosting of attractor tuning. We further compare visual pinging with TMS perturbations, and we find qualitative differences suggesting different underlying mechanisms. Based on the difference in behavioral impact of these 2 perturbation protocols (visual pinging does not affect working memory behavior, but TMS does), we speculate that visual pinging may increase EEG decodability via reduced across-trial variability or by transient boosting of active attractors, while TMS-induced reactivations would be supported by activity-silent mechanisms. Note that temporarily boosting an active attractor should not have a strong impact on behavior beyond the boosting period (unless additional long-lasting cellular mechanisms are engaged), while true reactivations from activity-silent stores should have a long-lasting impact, as the silent trace is refreshed.

We show here that relevant memories are stored in active codes and are thus decodable, while irrelevant memories could not be decoded, therefore potentially discarded. This interpretation sheds light on the intriguing ineffectiveness of pinging on discarded memories, while being effective in "reactivating" unattended but still relevant ones [7]. The reactivation effectiveness of pinging appears now to depend on whether memories are maintained in active neuronal representations, decodable from EEG. In line with this, a recent study shows that pinging reveals diffusing dynamics [45], a hallmark of active memories [45,49], instead of decaying dynamics, as expected in the activity-silent framework [1,45]. Our data reconcile the influential study by Wolff and colleagues [7] with recent works showing irrelevant or unattended memories actively encoded in scalp EEG [32–34], in the activity of cortical association areas using large-sample fMRI analyses [30] or intracranial recordings in monkeys [22,50], and in neural activity in visual areas of rodents [36]. While there is extensive evidence for long-lasting cellular and synaptic mechanisms in cortical neurons ("silent" mechanisms, e.g., [51]) that must coexist [52] with, or even support [17,40,53–55], active representations such as those reported in this study, there is more scant evidence that working memory can be volitionally stored without spiking activity [1,56].

Explicit evidence for activity-silent processes is difficult to obtain but is particularly confounded in the presence of active representations. A possible approach is to seek evidence for activity-silent traces from previously memorized but already irrelevant items [5,57], for which chance-decoding is expected in principle. There, too, it is hard to discard low-powered designs, so positive evidence must be sought. Recently, it has been reported that, between consecutive trials—when the previous memory should be discarded, similarly to uncued memories in [7]—neurons fire more synchronously after having been engaged in active working memory storage [5], suggesting that discarded memories can leave involuntary silent traces. Importantly, selecting sessions based on good overall decodability (as in Fig 3) did not reveal

decodability of these across-trial discarded memories (Fig 1 in [5]), supporting a true activity-silent substrate. We argue that the effect of induced reactivations must be validated during similar conditions in which memories are demonstrably discarded, presumably leaving an activity-silent trace. Current evidence shows across-trial behavioral effects of TMS, as is expected by reactivating previous, discarded memories [5,44], but not of visual pinging during intertrial periods (chapter 4 of [46]). Also, the evidence in Fig 3 suggests that visual pinging cannot reactivate putative activity-silent traces following a discarded item.

Finally, our results suggest that voltage and alpha power encode similar working memory content. Previous work, however, shows that alpha and raw voltage play different roles in working memory [25–27], in particular that alpha power tracks spatial attention instead of the actual memory content. In principle, with the experimental design of Wolff and colleagues [7], subjects could recode orientation as a spatial location, which would be tracked by alpha power and by sustained voltages [25]. Future work with other experimental designs including independent variation of orientation and attentional location [25] could clarify this point further. Regardless of whether EEG is tracking orientation or a location recodification, we argue that in these data, both signals are carrying analogous contents, with voltage being noisier.

In sum, our results add to previous literature showing robust decoding of unattended working memories from electrophysiological signals [22,30,32–36,50]. Our analyses reinforce the idea that interpreting null decoding as evidence for storage in silent traces is not straightforward, because null results might result from weak signals in insufficiently powered analyses [30,31].

## Methods

### EEG experiments

We analyzed 2 available datasets of visual pinging [7,18]. For decoding and EEG variability analyses, we focused on both experiments from [7]. In experiment 1 ($n = 30$), subjects were cued for which item was going to be probed (cued item, here also called attended, or uncued item, here called discarded memory). In experiment 2 ($n = 19$), subjects had to alternate their attention between 2 items (their early/late, here attended/unattended memory item). Experiment 1 consisted of 1 session, while experiment 2 consisted of 2 sessions (separated by approximately 1 to 2 weeks) on the same set of subjects. For variability analyses, we also analyzed the experiment of [18] ($n = 24$). In this experiment, the subjects had to memorize 1 item, thus always within the focus of attention. Importantly, the item decodability from raw voltage never dropped to chance. Additionally, we also analyze the voltage variance of the experiment 2 ($n = 6$) of a TMS study [6]. We refer the reader to the original studies for extra details [6,7,18]. All these datasets were made available in a fully anonymized format and had been approved by the corresponding institutional review boards, as indicated in the original publications.

### Data preprocessing

The data available online was epoched and baselined relative to the beginning of each epoch (S3 Fig). To revert this baselining, we computed trial-by-trial voltage difference between consecutive epochs. We then added this voltage difference to the beginning of each baselined epoch, effectively reverting all baselining effects. Additionally, for the variability analyses (see below), we remove any signal drift caused by, e.g., moving electrodes using the python function scipy. *signal.detrend* on each subject's variability. Finally, for the decoding analysis (see below), we also computed the alpha power. For this purpose, the data were Hilbert-transformed (using the FieldTrip function "ft_freqanalysis.m") to extract frequencies in the alpha-band (8 to 12 Hz), and total power was calculated as the squared complex magnitude of the signal.

## Decoding analyses

We used freely available code to perform these analyses, so we will only briefly describe the methodology here. For a detailed description of the decoding methods, please refer to the original study [7]. As in the original study, we decoded from all the 17 posterior channels (P7, P5, P3, P1, PO7, PO3, and O1 versus P8, P5, P6, P4, P2, PO8, PO4, and O2). Briefly, we collected the Mahalanobis distance between all possible pairwise combinations of the orientations and thus form a representational dissimilarity function. Finally, the decoding strength was calculated as the vector strength of this function. We decoded from raw voltage or alpha power with the exact same code. The decoding strengths (Fig 1) were smoothed over time with a gaussian kernel (SD = 10 ms).

## Across-trial variability analyses

We computed variability as the variance (var) across trials of the raw voltage traces (trials × sensors × time). Before averaging variances across sensors, we detrended them using the function *scipy.signal.detrend* to account for any drift in the signal. Finally, we computed the percentage of variability change (Δvar) relative to the baseline period of 2 s before the pinging stimulus (*b*): Δvar = (var − b) / b * 100. This referencing to the baseline ensures that changes in variability can be attributed solely to the pinging and not to other factors that are common to both pre and after pinging, such as varying stimulus orientations. We computed across-trial variability for each session separately, and then averaged variances across sessions. This is important for the TMS experiment, in which TMS location was held fixed. This way, our variance analysis is not capturing aspects of the design that vary from session to session (e.g., spatial attention, TMS target location).

## Fano factor

To compute the variability drop in the simulated spiking activity, we used the Fano factor [38], which is defined as the variance of spike counts in a given window (100 ms) divided by their mean. We then computed ΔFF, as the difference relative to the baseline period of 2 s before pinging stimulus.

## Phenomenological simulations of EEG trials

To study how single-trial baseline correction impacts pinging-induced increases in EEG decodability, we applied our decoders to synthetic EEG data generated by a model where spurious EEG reactivations are caused by a reduction in noise variability. First, we simulated 2 hypothetical delay maintenance EEG time series representing 2 independent experimental conditions (i.e., grating oriented $0^0$ versus $45^0$; $n = 200$ each condition) using the following Gamma function (f) as a single-trial waveform:

$$f(t|a, b) = \frac{e^{a-1}}{[b(a-1)]^{a-1}} t^{a-1} e^{-t/b}$$

For each trial and condition, parameters $a$ and $b$ were drawn from a gaussian distribution (condition 1, $\mu_a = 2$, $\mu_b = 130$ ms; condition 2, $\mu_a = 3$, $\mu_b = 80$ ms; all conditions with $\sigma_a = 0.2$, $\sigma_b = 0.5$ ms). Each waveform was then scaled by 0.5 and 0.25, respectively, and the time onset was set to time point 0.1 s.

Finally, for each single-trial and condition, gaussian noise was added ($\mu = 0$, $\sigma = 0.75$). During the impulse period (1.3 s to 1.5 s), for each trial and experimental condition, we reduced 30% the variability of the noise ($\sigma = 0.525$).

To use the same decoding method as the original publication, we generated a multichannel time series. We created 2 sets of dipoles located around left (position [1.5–8.6 1.5] cm, orientation [−1 −1 −1]) and right (position [1.5–8.6 1.5] cm, orientation [1 −1 −1]) primary visual cortex, and the simulated time courses were projected to the scalp via a forward model [58]. S3C Fig plots decoding of these signals upon different conditions of EEG voltage baselining. The Matlab code for these simulations is available on https://github.com/comptelab/reactivations.

## Activity-silent network model

We used a previously proposed computational model [5] to simulate memory reactivations. The model consists of a network of interconnected 2,048 excitatory and 512 inhibitory leaky integrate-and-fire neurons [59]. This network was organized according to a ring structure: Excitatory and inhibitory neurons were spatially distributed on a ring so that nearby neurons encoded nearby spatial locations [60]. Excitatory connections were all to all and spatially tuned, so that nearby neurons with similar preferred directions had stronger than average connections, while distant neurons had weaker connections. All inhibitory connections were all to all and untuned. Network parameters were taken from [5]. Simulation of "activity-silent" mechanisms was done by simulating 2 presynaptic variables x and u, as described in [5]. Reactivations were accomplished stimulating all excitatory neurons with a nonspecific external stimulus [5].

## Attractor-boost network model

For the attractor-boost model, we used a bump attractor model similar to the activity-silent model described above, but without short-term plasticity. As in this other model, attractor boosting was achieved with a nonspecific external stimulus to all excitatory neurons. The Brian [61] code for this model is available on https://github.com/comptelab/reactivations.

## Improving statistical power

**Smoothing.** To improve signal-to-noise ratio for our decoding analyses, we smoothed the voltage traces using a gaussian kernel with σ = 32 ms. This was in contrast with the original study that used instantaneous, nonsmoothed voltages for decoding and smoothed the resulting decoding accuracies with σ = 16 ms.

**Pooling all trials, across different sessions.** We also pooled all trials across sessions and subjects. Because all subjects and sessions consisted of a similar number of trials, we are not biasing our analyses toward a specific subject. Note that we only pooled trials across sessions for the analyses in Fig 2. We averaged across sessions for the other analyses.

**Cross-validated median split.** To simulate an increase of signal-to-noise ratio, we removed sessions with low decodability. Importantly, to avoid circularity in our analysis, we cross-validated this selection in the following way. For each session, we randomly split the single-trial decoding accuracies in two-halves. With the first half, we sorted the sessions by their average decoding accuracy during early delay [0.25 to 0.45] s, and we selected high and low decoding sessions (median split). We then computed the average decoding accuracy in the second half for low- and high-decoding sessions defined in the first half. We repeated this procedure 2,000 times for different random half-splits of the data (split-folds). Finally, we established the chance-level decoding accuracy (shuffle predictor) for high-decoding sessions by averaging the decoding strengths of 1,000 shuffled decoders (i.e. permuting orientation labels) analyzed with the same procedure (200 split-folds). For high-decoding sessions, we considered the average decoding strength significant if at least 95% of the split-folds were higher than the shuffle predictor.

## Supporting information

**S1 Fig. A sustained alpha-power code tracks behaviorally relevant orientation memories.** Data from Wolff and colleagues (2017) [7]. (TIF)

**S2 Fig. In an attractor-boost network model, a nonspecific stimulus increases tuning without reactivation.** Two example stimulations of a bump-attractor with (weak drive) and without (no drive) a nonspecific drive at the end of the trial. Importantly, we did not include short-term plasticity in either simulation; thus, reactivations are not possible. Right, tuning during the last 0.5 s is higher for the trials in which a nonspecific drive was delivered (green), compared to when no drive was delivered (black). Similar models have been put forward in previous publications [29] (see also [39]). (TIF)

**S3 Fig. Effect of EEG baselining on stimulus decoding analyses.** (A) Original data from Wolff and colleagues (2017) [7] was baselined twice. (B) For our analyses, we de-baselined the second baselining of each trial so we could get continuity in EEG voltage traces through the whole trial (early baseline). Importantly, the exact time of the baseline affects the strength of "reactivations." Note that data with early baselining do not show any visible increase in EEG decoding. (C) Top: diagram outlining the computer simulation that generated the EEG synthetic data, as if 2 current dipoles were placed within the visual cortex (yellow) to recreate artificial trials. Two different sets of trials ($n$ = 200, blue and red), corresponding to 2 different stimuli (e.g., 0˚, 45˚) were generated (Methods). We simulated 2 event-related potentials (electrode "Oz"; white) followed by a drop in across-trial variability. Bottom: illustration of how a similar signal-to-noise ratio increase (shown in "decoding" for the drop in variance case) can result from a drop in variance or actual reactivation (signal increase). (D) Through simulations, we show that the baselining procedure introduces spurious reactivations in data without any true reactivation signal. By applying the decoding methods of Wolff and colleagues [7], we observe that (as in the data) spurious reactivations are barely visible with a distant baseline (early baseline) but are amplified for more proximal baselines (late baseline). These analyses illustrate that baselining is problematic and should be avoided, as it has been previously pointed out [28]. Darker lines mark the impulse period in the data and drop in variance in the simulations. Data from Wolff and colleagues (2017) [7]. (TIF)

## Acknowledgments

We thank Nathan Rose for sharing part of Rose and colleagues (2016) [6] dataset privately and the authors of Wolff and colleagues (2015, 2017) [7,18] for sharing the full dataset publicly. We thank Michael Wolff and Mark Stokes for constructive discussions.

## Author Contributions

**Conceptualization:** Joao Barbosa, Diego Lozano-Soldevilla, Albert Compte.

**Data curation:** Diego Lozano-Soldevilla.

**Formal analysis:** Joao Barbosa, Diego Lozano-Soldevilla.

**Funding acquisition:** Joao Barbosa, Albert Compte.

**Investigation:** Joao Barbosa, Diego Lozano-Soldevilla, Albert Compte.

**Methodology:** Joao Barbosa, Diego Lozano-Soldevilla, Albert Compte.

**Software:** Joao Barbosa, Diego Lozano-Soldevilla.

**Supervision:** Albert Compte.

**Writing – original draft:** Joao Barbosa.

**Writing – review & editing:** Joao Barbosa, Albert Compte.

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
