## [Editor Report · Decision Letter 0]

2 Apr 2021

Dear Albert, 

Thank you for submitting your manuscript entitled "Unattended short-term memories are maintained in active neural representations" for consideration as a Short Report by PLOS Biology.

Your manuscript has now been evaluated by the PLOS Biology editorial staff, as well as by an academic editor with relevant expertise, and I am writing to let you know that we would like to send your submission out for external peer review.

Please re-submit your manuscript within two working days, i.e. by Apr 06 2021 11:59PM.

Kind regards,

Gabriel Gasque, Ph.D.,

Senior Editor

PLOS Biology

---

## [Decision Letter · Decision Letter 1]

6 May 2021

Dear Albert,

Thank you very much for submitting your manuscript "Unattended short-term memories are maintained in active neural representations" for consideration as a Short Report at PLOS Biology. Your manuscript has been evaluated by the PLOS Biology editors, by an Academic Editor with relevant expertise, and by three independent reviewers. You will note that reviewers 1 and 3 have revealed their identities. 

In light of the reviews (below), we will not be able to accept the current version of the manuscript, but we would welcome re-submission of a revised version that takes into account the reviewers' comments. We cannot make any decision about publication until we have seen the revised manuscript and your response to the reviewers' comments. Your revised manuscript is also likely to be sent for further evaluation by the reviewers.

We expect to receive your revised manuscript within 3 months. 

**IMPORTANT - SUBMITTING YOUR REVISION**

Your revisions should address the specific points made by each reviewer. As you will see, all reviewers agree the work is timely and important, and most of their concerns can be addressed with textual changes --we think-- regarding issues of interpretation and providing appropriate context and discussion of other related work. In addition, reviewers 2 and 3 also suggest some additional analyses that could bolster your conclusions. Having discussed these comments with the academic editor, we think it is important that you include those new analyses in the current manuscript.

Please submit the following files along with your revised manuscript:

*Re-submission Checklist*

*Published Peer Review*

*PLOS Data Policy*

*Blot and Gel Data Policy*

Sincerely,

Gabriel Gasque

Senior Editor

PLOS Biology

ggasque@plos.org

REVIEWS:

Reviewer #1, Brad Postle: "Unattended short-term memories are maintained in active neural representations," Barbosa, Lozano-Soldevilla, and Compte. This manuscript present the reanalysis of data from two publications (from other groups) about which, as the authors write, "The[ir] interpretation of reactivations from absent decoding as evidence for 'activity-silent' storage has had a strong impact in the field." The major findings are threefold. First, reanalysis of the data from Wolfe et al. (2017) replicated the original finding of failing to decode evidence for an active representation of the unprioritized memory item (UMI) when the EEG data were analyzed in their voltage format, but SUCCEEDED in decoding evidence for an active representation of the UMI from the alpha-band component of the signal, after these same data had been spectrally transformed. Second, they show with simulations, then empirically, that the effect of the visual "ping" is to decrease trial-to-trial variability in the signal. Third, they show with simulations, then with empirical analysis of the data from Rose et al., (2016), that the effect of a pulse of TMS is the opposite of that of a ping: TMS increases trial-to-trial variability. The significance of (2) and (3) is that (2) may explain why the effect of the ping on time-voltage data is to briefly rescue decoding of the UMI, whereas (3) would seem to indicate that the activity-silent interpretation of TMS-evoked rescue of decoding of the UMI remains viable (although, as this review will highlight, this interpretation is equivocal in the manuscript as currently written). This manuscript will be of considerable interest to many who study working memory from a variety of perspectives (computation, behavior, intra- and extracranial electrophysiology, fMRI, neurostimulation…), because of its important implications at both empirical and theoretical levels. (As the senior author of Rose et al. (2016) I can't not forgo anonymity and still write a comprehensive review.)

This work is important and will get a lot of attention, but I worry that some of the framing fails to emphasize some of the most important elements and pushes a narrative that's both too strong and somewhat out of date. Let's start with the title - it's stating proposition that has already been described on several occasions, including in one paper that they cite, but not in this context (Christophel et al. 2018), and several that they don't (van Loon et al. 2018, Wan et al. 2020, Yu et al. 2020, Libby and Buschman 2021). This latter group all document a phenomenon that the decoding methods used here can't distinguish, but that the authors might consider investigating with a different method, which is that the active representation of the UMI is transformed from its format when it is a prioritized (P)MI. (Two of these papers are from my group, and I want to be clear that I have no expectations that the authors need to cite them.) Furthermore there's an implication here and in many parts of the manuscript that the UMI in the Wolfe et al. studies is NOT also stored in an activity-silent format, but of course the authors can't know that. (Finally, with regard to the title, the authors of both (Wolff et al. 2017) and (Rose et al. 2016) refer to their tasks as "working memory" tasks, and the authors might consider using this label, which is much more widely used in the field.) This false dichotomy (active vs. activity-silent) is particularly dissonant in the final paragraph, in which the authors seem to ignore their own (very elegant) work that showed compelling positive evidence for activity-silent representations in the PFC of the monkey. Also important to consider are different results from the monkey PFC in which a "cognitive ping" reveals otherwise undetectable (and so 'probably' activity-silent) representations of stimuli (Stokes et al. 2013). Finally, as the authors (seemingly grudgingly) acknowledge, the most straightforward conclusion from their reanalysis of the data from Rose et al. (2016) is the UMI in that study most likely was held via activity-silent mechanism. Finally, it need not be the case that the idea of activity-silent representation "stands in contradiction with computational models of 'activity-silent' storage, where short-term plasticity changes are induced by neuronal activity" if one allows for existence of cognitive control. Indeed, Jacqueline Fulvio in my group has show behaviorally that the 'behavioral reactivation' TMS effect is subject to control (Fulvio and Postle 2020). (Again, I'm not asking for citations, but it is the case that I work in this area …) Masse et al. (Masse et al. 2019) and Manohar et al. (Manohar et al. 2019) have demonstrated that the two formats can, in principle, co-exist, and if the authors embraced this idea they could preempt readers like me getting distracted by an issue that's not the main point of the results.

I realize that I spilled a lot of ink to make this point, but it is really the only big-ish concern that I have with this otherwise excellent paper. I'll make more specific comments in the order in which they appear.

2nd paragraph: Rose el al. (2016) also reported an inability to decode the UMI from spectrally transformed data (Figure S5, which also show that the PMI is decodable from alpha and the UMI reactivation from beta).

Results and Discussion: The first time the key result is mentioned "We found that a sustained alpha power code tracks the items that remain relevant for future behavior" it'd be helpful to specify that it's tracking orientation; during my first read-through I was uncertain whether it was orientation or location-on-the-screen that was being decoded.

"Furthermore, it challenges the current view on the role of alpha power during working memory maintenance [18], which would suppress immediately irrelevant memories [19]." This is unclear, because 18 argues against a suppression/inhibition for alpha?

This clause needs more unpacking "possibly reflecting the prioritization of strong competition between actively held memories in attractor networks [20,21]" because someone unfamiliar with the details of 20, 21 won't necessarily understand what is meant here.

Is the work of Bae and Luck (recovery of decoding of previous trial's stimulus during the next trial) also relevant here, perhaps as part of a more detailed consideration of the implications of (Barbosa et al. 2020)?

Data preprocessing: it's unclear what is meant by the words "to revert this baselining." Additionally, a sentence or two explaining in more detail how this was done, and why it was important to do, would be helpful.

"funcion" is not English.

Figure 1: this is picky, but "decoding from alpha power (Methods), which reveals a strong sustained code of the unattended stimulus" strikes me as imprecise. What's being revealed is a stimulus representation, and from that one infers that there is a code that supports this representation, right?

Figure 2: "A signal-to-noise ratio increase can reflect a drop in variance …" Isn't it meant that an SNR increase can result from a drop in variance?

Signed, Brad Postle

Barbosa, J., H. Stein, R. L. Martinez, A. Galan-Gadea, S. Li, J. Dalmau, K. C. S. Adam, J. Valls-Solé, C. Constantinidis and A. Compte (2020). "Interplay between persistent activity and activity-silent dynamics in the prefrontal cortex underlies serial biases in working memory." Nature Neuroscience 23: 1016-1024.

Christophel, T. B., P. Iamshchinina, C. Yan, C. Allefeld and J.-D. Haynes (2018). "Cortical specialization for attended versus unattended working memory." Nature Neuroscience 21: 494-496.

Fulvio, J. M. and B. R. Postle (2020). "Cognitive control, not time, determines the status of items in working memory." Journal of Cognition 3: 1-8.

Libby, A. and T. J. Buschman (2021). "Rotational dynamics reduce interference between sensory and memory representations." Nature Neuroscience.

Manohar, S. G., N. Zokaei, S. J. Fallon, T. P. Vogels and M. Husain (2019). "Neural mechanisms of attending to items in working memory." Neuroscience and Biobehavioral Reviews 101: 1-12.

Masse, N. Y., G. R. Yang, H. F. Song, X.-J. Wang and D. J. Freedman (2019). "Circuit mechanisms for the maintenance and manipulation of information in working memory." Nature Neuroscience 22: 1159-1167.

Rose, N., J. J. Larocque, A. C. Riggall, O. Gosseries, M. J. Starrett, E. Meyering and B. R. Postle (2016). "Reactivation of latent working memories with transcranial magnetic stimulation." Science 354: 1136-1139.

Stokes, M. G., M. Kusunoki, N. Sigala, H. Nili, D. Gaffan and J. Duncan (2013). "Dynamic coding for cognitive control in prefrontal cortex." Neuron 78(2): 364-375.

van Loon, A. M., K. Olmos-Solis, J. J. Fahrenfort and C. N. L. Olivers (2018). "Current and future goals are represented in opposite patterns in object-selective cortex." eLife 7: e38677.

Wan, Q., Y. Cai, J. Samaha and B. R. Postle (2020). "Tracking stimulus representation across a 2-back visual working memory task." Royal Society Open Science 7: 190228 

Wolff, M. J., J. Jochim, E. G. Akyürek and M. G. Stokes (2017). "Dynamic hidden states underlying working-memory-guided behavior." Nature Neuroscience.

Yu, Q., C. Teng and B. R. Postle (2020). "Different states of priority recruit different neural codes in visual working memory." PLoS Biology 18: e3000769.

Reviewer #2: Summary: The authors investigate whether they can decode information that was previously thought to be maintained in silent working memory. The authors ask this question by using alpha power at posterior electrodes to decode working memory representations. In at least one experiment, they are able to decode working memory information. Furthermore, the authors then suggest that visual "pinging" may change the signal-to-noise ratio of EEG activity, thus increasing decoding accuracy of already active neural representations (by increasing SNR), rather than re-activating latent traces. Many have pointed out already that absence of evidence is weak evidence for absence of decodable memory activity, and this study is an important example of this point. Given the strong influence of the "activity silent" models of working memory storage, the present findings provide a critical alternative explanation for one of the more prominent studies arguing in favor of activity-silent modes of storage in working memory. The present authors also report simulations that support their hypothesis regarding the SNR effects of visual "pinging" (i.e., presentation of an irrelevant visual stimuli), showing that pinging may reduce across trial variability and thus increase SNR. This adds strength to the authors' speculations that the putative "silent" representations may have simply been masked by noise rather than truly silent. 

However, the authors' account doesn't provide a direct explanation for why it was the *relevant* and not the irrelevant memory representation that was decodable in the raw voltages after visual pinging. That is, while it is clear that they have successfully decoded using alpha power, why would visual pinging only resurrect the relevant item's representation if the authors are correct about the effects of pinging on across trial variability? Those variability effects should influence decoding of both relevant and irrelevant items, shouldn't they? 

The authors assert that "…alpha power tightly tracks working memory contents, *regardless* of their immediate behavioral relevance." (p. 3, results and discussion), but I thought figure 1 was ambiguous on this point. 1C shows a trend towards higher decoding strength for the relevant item, especially at the end of the delay period. Figure 1D also seems to show a divergence of the relevant and irrelevant decoding strengths at 1sec, but I wasn't sure why the irrelevant item line was dashed instead of solid (perhaps this is indicated somewhere in the manuscript, but I couldn't find it. I'd recommend making this more clear in the figure caption). 

This then leaves the mystery of why only the relevant item appears to be tracked by EEG voltage following the ping. The authors do state, "However, while attended memories are decodable both in alpha power and voltage traces, unattended memories are only detected in alpha power, *possibly reflecting the prioritization of strong competition between actively held memories in attractor networks.*" Does this sentence imply that the traces reflected in EEG voltage were indeed "silent" prior to the pinging? I thought this was unclear from the manuscript. The later arguments about pinging reducing across trial variability seem to leave open the possibility that both attended and unattended items were represented in EEG voltage, but with SNR too low to confirm it with the original analysis pipeline. I think this points merits careful clarification in the manuscript.

The authors also point out that TMS reactivation may be qualitatively different, because simulation suggest that TMS *increases* across trial variability, and that this falls in line with the predictions of "a biophysical network model of memory reactivation from silent, synaptic traces". I thought this was an interesting point, but it seems to sparsely described to really have a strong impact in the paper. Moreover, while being "potentially consistent" with that biophysical model is interesting, how strong is this evidence? I would be surprised if models that denied the role of "activity-silent" representations in working memory could not also be "consistent" with the finding that across trial variability is increased following TMS. But if the authors have a compelling argument that I should be surprised here, I think it should be clearly spelled out in the paper. If there is no strong argument, then I'd recommend tempering this particular conclusion. My view is that the really clear evidence in this study is the positive findings with alpha power in the pinging studies, and the simulations of how changes in SNR could explain the original findings in the pinging study. The other speculations regarding the potential causes of the TMS findings are not yet as convincing. 

Minor points: 

1. The authors use the same analysis pipeline as the originally published papers, with the inclusion of alpha power instead of raw EEG amplitude. This is a powerful approach. However, the original analysis pipeline only included 17 posterior electrodes. It is possible that information about working memory could also be represented in other electrodes on the scalp. Therefore, decoding accuracy could actually be higher if the authors deviated from the original methods and included all electrodes in their models. It may be useful to include an additional analysis that addresses whether a model that includes all electrodes increases decoding accuracy of working memory representations. 

2. In the current manuscript, the authors clearly show that alpha power tracks working memory representations in the time periods where the prior work suggested that there were no neurally active representations. This is the critical empirical pattern. That said, recent work suggests that alpha power and EEG voltage may play qualitatively different roles in working memory tasks:

Bae, G. Y., & Luck, S. J. (2018). Dissociable decoding of spatial attention and working memory from EEG oscillations and sustained potentials. Journal of Neuroscience, 38(2), 409-422.

Hakim, N., Adam, K. C., Gunseli, E., Awh, E., & Vogel, E. K. (2019). Dissecting the neural focus of attention reveals distinct processes for spatial attention and object-based storage in visual working memory. Psychological Science, 30(4), 526-540.

So, it may be worthy of some discussion whether the active neural signal the present authors have identified might reflect a different aspect of working memory maintenance than do the patterns of activity in EEG voltage. This could be an interesting compromise between the original framing of the prior reactivation studies and the present framing. 

3. The published data was baselined, and the authors reversed this baselining for their analyses. The authors should provide further explanation for why they reversed this baseline and should additionally discuss whether they were able to decode working memory representations with the published, baselined data. 

4. The authors introduce fano-factors in Figure 3, but do not include a discussion of fano-factor anywhere else in the paper. Given the prominence of "△fano-factor" in their figure, the authors should include a brief description of fano-factors in the Method's section of their manuscript. This description will help bridge the literature that typically uses fano-factors to the literature that typically investigates alpha power activity in human EEG. 

Conclusions: 

The authors make two main conclusions (1) working memory representations that were previously thought to be maintained in an activity silent state have been shown to be tracked via an active neural trace in the alpha band, and (2) visual "pinging" changes the ratio of signal-to-noise in EEG signals, thereby allowing active signals that may have been masked by noise to be detected. This work provides an important new interpretation of a highly influential set of studies, and I believe it would have a strong positive impact in the literature that will generate vigorous follow-up work. Their conclusions, however, would be strengthened by addressing the above comments. 

Reviewer #3, Thomas Christophel and Vivien Chopurian: The authors report a short reanalysis of EEG data from recent studies investigating reactivation (via 'pinging') of supposedly 'activity-silent' mnemonic traces during working memory. This reanalysis shows that alpha band activity carries an active trace of memorized content, in contrast to this prior work. Additional reanalyses concern the nature of the 'pinging' effects reported in this and other prior work. The manuscript argues for differential effects of 'pinging' using either sensory mask-like stimuli and TMS on the reduction and increase of across-trial EEG variability, respectively.

The finding that data in a core study seemingly supporting 'silent' working memory contains evidence for active neural representations alone is critical to the field and beyond. This finding is robust and consistent with previous work and without any doubt deserves the attention given here. There are opportunities to elaborate more on this finding and its relationship to other work, but it is convincing as is. The additional reanalyses and their interpretations highlight some explanations for 'pinging' effects, but several concerns let me doubt the conclusions drawn here. 

We will outline our suggestions, concerns, and recommendations in the following sections. Beforehand, we would like to emphasize that our expertise primarily lies in related fMRI work. 

Major:

* Our concerns solely relate to the second part of the manuscript which argues that "visual pinging reveals an underlying active code by quenching EEG noise". As the authors outline this is indeed a plausible explanation, but it is questionable whether the data analyses reported are sufficient to support the hypothesis in a substantive way. 

As the authors outline, a reduction in trial-by-trial variability is the expected outcome of introducing a stimulus into a system that is constant across trials. Simply put, stimulation (like a 'ping') can be seen to replace endogenous by with exogenous activity. If this exogenous activity is constant across trials, variability across trials is reduced. It is equally plausible that in a decoding analysis, the response related to a memorized item is more easily decoded when noise is reduced. The 'noise' in this decoding analysis and the 'trial-by-trial variability' in the analysis provided, however, are not the same. On the contrary, the trial-by-trial variability is composed of both the noise and the signal used for the decoding analysis. This becomes obvious as one considers the fact that subjects memorize different orientations on different trials, making the signal itself vary across trials. How different components (ping signal, mnemonic signal, and residual noise) are aggregated to form the overall EEG signal is unknown, but here we see little indication that trial-by-trial variability can be used as an exclusive indicator of the residual noise component. For these reasons, we do not see how the trial-by-trial variability analyses provide evidence in favor or against the 'quenching noise' hypothesis.

* This problem becomes even clearer when considering the trial-by-trial variability in Rose et al: In Rose, not only does the content vary across trials, but the ping is variable across trials, too (in the EEG Exp 2). Considering that this means that on different trials the TMS coil is pointed at different parts of the brain, one might expect to see it in the absence of any mnemonic or cognitive activity (e.g. in a lifeless neuronal substrate). Notably, this also provides a relatively straight forward explanation for the 'reactivation' in Rose et al. (in their Exp 2, at least): The TMS itself already carries information about the memorized content, which makes decoding the content after the pulse a trivial finding. 

* What It is essential here,, is that a critical finding (active representations in the Wolff data) is not hampered in impact, by jumping to conclusions in the second part of the manuscript. While the 'quenching noise' interpretation is a plausible one, alternative interpretations need to be considered. One simple interpretation is that 'pings' increase overall alertness or modality-specific attention which in turn increases the signal of any memorized content. This could be seen as a mechanism to counter interference by distracting stimulation (see Bettencourt and Xu as well as Rademaker et al., in particular, the difference between expected and unexpected distraction in Bettencourt and Xu). This highlights that a robust explanation of 'pinging' effects must consider both enhancing ('pinging') and disrupting ('distracting') effects of stimulus presentations during working memory delays which are likely to jointly affect neural decoding in different ways across studies. 

Minor:

* The main finding of the paper is a rather simple but important one: If you run the same analysis as Wolff et al. (2017) with the same data but using Alpha power rather than raw data, then you can decode unattended items, which were not decodable before. One potential avenue to make this manuscript even more insightful would be to provide further analyses that help explaining this finding.

As the authors mention, raw EEG power already carries information about a memorized item in Wolff et al. (2015) when combining 'pinging' and 'no-pinging' trials prior to the 'ping'. The same analyses split half for 'pinging' trials results in an occasional null result ('long trials only'). This highlights the trivial but apparently forgotten idea that statistical power might be the essential determinant of whether one is likely to find a positive result in a given study. The change from raw signal to Alpha power might simply constitute an increase in per trial effect size (like other forms of filtering and smoothing). In other words, the question is whether the difference between raw and Alpha -based analyses represents a qualitative difference in the underlying signals (between UMI and AMI) or simply a quantitative difference in the strength of representation.

We suggest quantifying this effect by running raw-EEG and Alpha decoding for both studies and 'simulating' studies with different sample sizes (by randomly removing subjects from a given study) or even different number of trials (e.g. split half). We also suggest running analyses combining decoding accuracy across time-points (e.g. averaging across the delay) on a given trial and/or combining trials prior to the decoding analyses (similar to run-wise beta decoding in fMRI) to increase statistical power for raw analyses. We want to stress that these are suggestions to better explain an already relevant findings rather than analyses necessary to support the finding. Not all these suggestions might be feasible for the datasets available. 

* One final recommendation is to investigate the differential involvement of different electrode positions in the decoding of AMIs and UMIs. We want to be very clear, however, that is more driven by the reviewers' curiosity than anything else. 

* "[…] moving electrodes using the python function signal.detrend on each subject variability." We assume this refers to the function in the scipy package and the linear detrending option thereof, please specify. There also might be a typo here ("each subject's").

* "Mahalanobis distance between all possible pairwise combinations of the orientations and thus form a tuning curve." Please specify how these pairwise distances are turned into a tuning curve. An array of pairwise distances is typically referred to as a representational dissimilarity matrix.

* "We realized that an earlier study had reported that attended spatial memories were decoded more reliably from EEG alpha power than from raw voltages [16]." This might be related to our primary expertise in fMRI data, but when reading Foster et al. (2016), our understanding is that they differentiate between evoked and total power not raw voltages and Alpha power. See David et al. (2006): "In short, evoked responses can be characterized as the power of the average; while induced responses are the average power that cannot be explained by the power of the average." 'Total' power would then be both components combined which (in our understanding) is what is used, here. 

Naturally, there is a relationship between evoked power and evoked responses in preprocessed (rather than raw) EEG data, so the results by Foster et al. still can serve as a motivation for the reanalysis in the current manuscript. The authors should however clarify what prior work found. 

* "Despite their relevance for upcoming memory-guided behavior, currently unattended memories cannot be robustly decoded from raw EEG voltage traces [6,7] (Figure 1a, red)." This might be somewhat misleading as prior work only showed that they were not able to decode from raw voltages which might simply be a false negative. The prior work does not show that UMIs "cannot" - in principle - be decoded.

* "Furthermore, it challenges the current view on the role of alpha power during working memory maintenance [18], which would suppress immediately irrelevant memories [19]." Here, it might be worth noting that suppression might lead to inverted tuning which could be decodable equally well than non-inverted tuning. This means that being able to decode an item does not necessarily mean that it isn't suppressed or that the signal one decodes isn't suppressive in nature. (See the different works by Lorenc and Postle and van Loon). Notably, there is plenty of debate about these inverted tuned representations, so it is unclear to me whether to include the debate here.

* "Despite apparent similarities, TMS reactivations impact working memory performance [5,6], while pinging does not [7]." Here it is worth noting that (by our count) in Rose et al. only one out of three TMS experiments show a behavioral effect of the TMS (one out of six possible effects in exp. 4 at p = 0.01, which is not reanalyzed in the current manuscript). Overall effects of interfering stimulation with distractor or TMS are rather scarce, so we caution to interpret this apparent difference more carefully.

* "Decoding from raw voltage or alpha power was done with the exact same code, but preprocessing the data differently.": The second sentence seems to miss a "with" or we suggest changing the voice from passive to active

* "Each 'trial' trace was simulated as a slope α, different for each stimulus (α1 = 1 and α1 = 1/2) on top of noise sampled from a normal distribution ξ, with mean 0 and standard deviation 1." 'α1 = ½' should probably be 'α2 = ½'. It would be helpful to explain the selection of these model parameters (why 1 and 1/2)? Further it might be helpful to spell out the simulation a little (e.g. What are the two items, what is the data being simulated, where does the data start, where is 0 ….), it took us more time than necessary to get what the authors are doing here and why. Arguably, however, the whole simulation can be seen as demonstrating the trivial fact that SNR means Signal to Noise ratio, but we'll leave it to the authors judgement whether explaining this is necessary. 

We wish you all the best for this fascinating project.

Thomas Christophel and Vivien Chopurian

---

## [Decision Letter · Decision Letter 2]

30 Aug 2021

Dear Albert,

Thank you for submitting your revised Short Report entitled "Pinging reveals active, not silent, working memories" for publication in PLOS Biology. I have now obtained advice from the original reviewers and have discussed their comments with the Academic Editor. 

Based on the reviews, we will probably accept this manuscript for publication, provided you satisfactorily address the remaining points raised by reviewer 3. Please also make sure to address the data and other policy-related requests listed below my signature.

We expect to receive your revised manuscript within two weeks. 

*Published Peer Review History*

*Early Version*

Sincerely,

Gabriel Gasque, Ph.D.,

Senior Editor,

ggasque@plos.org,

PLOS Biology

TITLE:

We would like to decompress your title a little, to make it more accesible and appealing to our broad readership. We recommend: 

Pinging the brain with visual impulses reveals an electrically active code for unattended working memories.

However, we would be happy to discuss alternatives, if you think our suggestion is inaccurate or misrepresents your findings.

BLURB:

Please provide a blurb, which will be included in our weekly and monthly Electronic Table of Contents, sent out to readers of PLOS Biology, and may be used to promote your article in social media. The blurb should be about 30-40 words long and is subject to editorial changes. It should, without exaggeration, entice people to read your manuscript. It should not be redundant with the title and should not contain acronyms or abbreviations.

DATA POLICY:

Please include in your Data Statement in the submission system an explicit statement on where the original data can be found (“the original publications.”), while keeping the following “All the code used for the analyses can be found at https://github.com/comptelab/reactivations”

Please also ensure that each figure legend in your manuscript includes information on where the underlying data can be found.

DATA NOT SHOWN:

Please note that per journal policy, we do not allow the mention of "data not shown", "personal communication", "manuscript in preparation" or other references to data that is not publicly available or contained within this manuscript. Please either remove mention of these data or provide figures presenting the results and the data underlying the figure(s).

Reviewer remarks:

Reviewer #1, Brad Postle: That authors have thoroughly and satisfactorily addressed all the points raised in my initial review.

Reviewer #2: I think the authors have done a strong job with the revision, and I believe the paper is ready for publication.

Reviewer #3, Thomas Christophel & Vivien Chopurian: The authors provide an extensive response to our comments. The revised manuscript includes several additional analyses and simulation which critically add to the prior work. The updated main finding shows that unattended memory items can be decoded from both raw and frequency transformed EEG data. Moreover, 'reactivations' by 'pinging' appear to depend on the presence of an active raw EEG trace of the unattended item. Finally, as an alternative explanation for pinging effects, they introduce an attractor-boost model which appear to equally explain their results. The authors have broadened their interpretation of their initial findings in the light of these additional findings and discuss possible underlying mechanisms.

These changes greatly improve the quality of an already noteworthy study. Some open points remain. These concerns predominantly relate to the interpretation of Rose et al., which play a lesser role in the revised manuscript. We will outline our suggestions, concerns, and recommendations in the following sections. 

Minor:

1. Data from Rose et al. (2016) is reanalyzed for the current study and plays a part in the conclusions drawn by the authors (e.g. in the discussion: 'We further compare visual pinging with TMS perturbations, and we find qualitative differences suggesting different underlying mechanisms'). The authors however did not analyze the TMS-EEG data in the same way as the Pinging data and conclude different mechanisms based on one analysis looking at difference in variation measures between the two sets of experiments. 

The authors now acknowledge that variation in the TMS site might result in an increase in variability but appear to contend that reactivations (meaning an interaction of the TMS with the respective conditions) are the more probable cause of this increase, as the TMS itself should average out due to the block-wise nature of their analysis. When looking at the data directly, however, we see that the by far largest increase in variability (so large that it evades the y-axis on the respective plot) occurs at 0ms relative to the pulse. In other words, the largest change in the EEG measurements is the pulse itself which - according to the authors - should have been averaged out. Hence, we would attribute the differences in variability found between pinging and TMS to variations in the TMS itself not a difference in the underlying mechanism of the reactivation which might be sensibly explained by boosted active representations or even by noise-quenching (which would be overshadowed TMS-related noise). We suggest a more careful interpretation of these differences in variability and possible underlying mechanisms.

2. For the across-trial variability analyses, please clarify the block-wise nature of this analysis in the methods section.

3. Please extend the y-axis for the variability analyses such that the reader can see the extend of the peak for the data from Rose et al. (2016). 

4. "We then tested the influence of sample size on decoding estimates in each condition, by varying the number of sessions and trials included in the analyses. This showed a striking difference between unattended and discarded memories: while increasing the sample size in the unattended condition resulted in a monotonic increase of t-value, it did not for discarded memories (see diagonal in Fig. 2). This result suggests that increasing the number of sessions would lead to decodability of unattended memories, but not of discarded memories." This monotonic increase might be less striking than one might think. In a bootstrapping procedure like this, the closer a randomly drawn sample of trials gets to the full set the closer it will approximate the results with all trials included. For unattended items this results appears to be p < 0.1 whereas for discarded it is p ~ 0.48, which appears to be the main difference here. Notably, p < 0.1 would constitute a significant result if tested using a one-sided test (p < 0.05) and we see little reason to perform a two-sided test (because below chance classification is neither hypothesized nor plausible unless confounds affect the cross-validation, see Görgen et al., 2018, Neuroimage). Please provide a detailed explanation of this analysis in the methods section. Furthermore, it is unclear to me what time-points are used for these plots (i.e., what p < 0.1 refers to). 

5. Throughout the manuscript the work by Panichello and Buschman (2021) is cited as evidence for active representations of unattended items found in intracranial recordings. In our view, this study cannot be easily interpreted as such evidence as (1) these items can be discarded after the cue and (2) these signals were only observed until 500 ms after the cue when they were in decline (i.e., there might have been too little time for them to reach baseline), At least in our reading, Panichello and Buschman might not agree with such a characterization of their results.

Best of Luck

Thomas Christophel & Vivien Chopurian

---

## [Editor Report · Decision Letter 3]

4 Oct 2021

Dear Albert,

On behalf of my colleagues and the Academic Editor, Frank Tong, I am pleased to say that we can in principle offer to publish your Short Report "Pinging the brain with visual impulses reveals electrically active, not activity-silent working memories" in PLOS Biology, provided you address any remaining formatting and reporting issues. These will be detailed in an email that will follow this letter and that you will usually receive within 2-3 business days, during which time no action is required from you. Please note that we will not be able to formally accept your manuscript and schedule it for publication until you have made the required changes.

PRESS

Sincerely, 

Gabriel Gasque, Ph.D. 

Senior Editor 

PLOS Biology

ggasque@plos.org